

# Solar cycle impacts on North Atlantic climate

Paula L.M. Gonzalez,[1,2] Lesley J. Gray,[1,3] Stergios Misios,[4,5] Scott Osprey,[1,3] Hedi Ma[6]

[1] National Centre for Atmospheric Science, UK

[2] University of Reading, Reading, United Kingdom

[3] Department of Physics, Oxford University, Oxford, United Kingdom

[4] Department of Geoscience, Aarhus University, Aarhus, Denmark

[5] National Observatory of Athens, Athens, Attiki, Greece

[6] Hubei Key Laboratory for Heavy Rain Monitoring and Warning Research, Institute of Heavy Rain, China Meteorological Administration, Wuhan, China

*Correspondence to*: Scott Osprey (scott.osprey@physics.ox.ac.uk)

**Abstract.** The North Atlantic sector has been identified as a region where the 11-year solar cycle has small but non-negligible impacts on near-surface climate. Nonetheless, debate persists about the robustness of such impacts and the mechanisms that explain them. The limited length of historical records, together with the complexity of separating the solar cycle influence from other forcings and from internal variability explain, at least partially, the discrepancies in published results. This work

explores the signatures of the 11-yr solar cycle over the North Atlantic in 20th Century reanalysis datasets, which provide longer reconstructions of climate by assimilating only long-record surface observations. The signatures are compared with those detected in long reconstructed observational datasets, modern reanalysis and previous studies. The results confirm previous studies and reveal a robust lagged boreal winter response in mean sea level pressure north of the Azores, peaking 2-3 years after solar maxima. The response is however non-stationary, varying both within the season and on multi-decadal

scales. An assessment of the impacts on sub-surface ocean temperatures using an ocean reanalysis dataset supports the hypothesis that thermal inertia of the ocean could explain the lag in the response and amplification of the response.  A re-emergence of warm anomalies in the years following solar maxima is observed over the central North Atlantic and is consistent with the observed solar response in sea level pressure north of the Azores.

## 1 Introduction

The 11-year solar cycle is characterized by a quasi-periodic oscillation in solar irradiance received at the top of the atmosphere (Lean and Rind, 2009). Variations in total solar irradiance (TSI) are dominated by energy in the visible part of the spectrum that can penetrate the atmosphere and reach the Earth's surface. The resulting impacts arrive through the so-called 'bottom-up' pathway, since its direct influence is on sea surface temperatures (SSTs) which can then indirectly influence patterns of weather and climate from below (Meehl et al., 2009; Gray et al., 2010: Misios and Schmidt, 2012). TSI variations over the 11-

yr solar cycle are associated with a global-average SST response of  approximately 0.1 K (White et al., 2003). As a result of



thermal inertia of the oceans the peak global SST response is generally found to lag the peak 11-year solar activity by approximately 1-2 years (White et al., 1997; Misios et al., 2016).

In addition, several indirect 'top-down' mechanisms have been proposed, for example through variations in the UV part of the solar radiation spectrum (see Gray et al. 2010 for a review) and/or through variations in energetic particles (Seppälä and
Clilverd, 2014; Maliniemi et al., 2019). Their effects can penetrate to the stratosphere and be amplified by ozone responses. Previous work highlighted that the top-down influences act mainly in winter and extend their influence downwards to the surface via wave-mean flow interaction, so that relatively small variations in the stratosphere can be amplified and transferred to the surface in the North Atlantic region (Kodera and Kuroda, 2002).

The 11-yr solar cycle has been previously linked to impacts in North Atlantic surface climate (e.g. Woollings et al., 2010; Gray
et al., 2010, 2013, 2016; Scaife et al., 2013; Thieblemont et al., 2015;  Kuroda et al., 2022). However, this region is influenced by many different forced variability and feedback processes and the robustness of the observed solar cycle signal has been challenged (Chiodo et al., 2019). Seasonal-to-interannual atmospheric variability in the North Atlantic sector is dominated by the North Atlantic Oscillation (NAO), particularly during boreal winter (Hurrell and Van Loon, 1997; Hurrell et al., 2003) and is also influenced by other global modes of variability including the Pacific-North-Atlantic wave-train, the Atlantic
Multidecadal Oscillation (AMO) and variability of the stratospheric polar vortex. The region also shows strong susceptibility to the influence of forced variability and feedbacks such as anthropogenic aerosols (e.g. Bellouin et al., 2020), volcanic activity (e.g. Swingedouw et al., 2017),  and arctic amplification (e.g. Barnes and Polvani, 2015). Interactions between these natural and forced sources of variability add to the climatic complexity of the region (Ottera et al., 2010; Dimdore-Miles et al., 2021; Klavans et al., 2022), so that solar signal detection is particularly challenging. Some climate model studies have simulated a
modulation of North Atlantic SSTs and mean sea level pressure (SLP) by the solar cycle but of weak amplitude (e.g. Ineson et al. 2011) and an improved predictability of NAO  by the solar cycle (Drews et al. 2022), although in many cases this is not distinguishable from unforced climate variability (Mitchell et al., 2015).

One reason that hampers a robust attribution is the short length of atmospheric observational records in relation to the 11-yr solar period, which means that relatively few cycles have been observed in sufficient detail. Another reason is that the
magnitude of the analysed solar cycle responses is very small in comparison to natural variability (Sjolte et al., 2018). In addition, detection and attribution of the surface solar response in the North Atlantic has been complicated by the fact that the observed response is lagged by several years. This was first identified by Gray et al. (2013) using the historical gridded HadSLP and HadISST datasets that span approximately 150 years. They identified a surface response that amplified and evolved for several years following the peak in solar forcing. The largest statistically significant response was located at the southern node
of the NAO, in the region of the Azores, with a lag of ~3 years. A response was also found over the northern NAO node, over Iceland, but was not statistically significant, possibly because natural variability is greater in that region.

A mechanism for the lagged surface response has been proposed that involves interaction of a forced NAO-like signal in SLP with the underlying mixed-layer ocean (Scaife et al., 2013) that can give rise to an amplification of the forced signal with a lag



of a quarter cycle (i.e. 3-4 years, as observed). This involves the formation of surface ocean temperature anomalies that, due
to the nature of the annual cycle in mixed-layer depth of the North Atlantic, can persist below the mixed-layer during the
summer months and then re-emerge at the beginning of the following winter. A further observational study, using the much
longer gridded SLP reconstruction dataset of Luterbacher et al. (2002) extending to ~350 years confirmed the long-term
presence of a lagged solar response over the North Atlantic (Gray et al. 2016) although the response amplitude varies over
time depending on the amplitude of the 11-yr solar forcing amplitude so that detection skill is likely to vary according to the
background state  (Ma et al. 2018).

Within this context, this work aims to revisit evidence for solar cycle signatures in the surface climate of the North Atlantic by
making use of reconstructed observational datasets and 20[th] Century reanalysis datasets. The reanalyses provide useful
validation of previous results, as well as providing improved statistics because they are produced using an ensemble approach.
In addition, we also analyse an ocean reanalysis to explore evidence for the re-emergence hypothesis and its consistency with
the atmospheric response.

The datasets and methodologies used in the study are described in Section 2. Section 3 presents the results of the study and
compares them to previous work. Section 4 provides a discussion of the results and outlines our conclusions.

## 2. Data and Methodology

### 2.1 Datasets

This study employs atmospheric and oceanic reanalyses and reconstructed observational datasets to examine the solar cycle
signatures in zonally-averaged zonal winds and temperatures, surface air temperature, sea level pressure and ocean potential
temperature. A summary of the datasets and variables is given in Table 1.

The ECMWF CERA-20C reanalyses (Laloyaux et al., 2018) is available for the period 1901-2010 and is produced using a
coupled atmosphere-ocean model that assimilates only conventional surface observations of the atmosphere and ocean. A 10-
member ensemble is available that accounts for uncertainty in the observational record as well as aspects of the underlying
model errors.

The NOAA-CIRES-DOE 20[th] Century Reanalysis version 3 (20CRv3) dataset is available for the period 1850-2014 and
similarly assimilates only surface pressure observations (Slivinski et al., 2019). This dataset was also built as an ensemble (80
members derived from 8 distinct SST initializations), but it is released as a best estimate together with an uncertainty measure.

The ECMWF ERA5 reanalyses over the 1950-2014 period (Hersbach et al., 2020) is used to examine and compare the response
signals over recent periods. The ERA5 dataset is produced by assimilating not only surface data but also an extensive array of
upper air meteorological observations. It therefore represents our best current estimate of the state of the atmosphere with an
improved representation of the stratospheric circulation, particularly after 1979 when satellite observations are introduced into
the assimilation procedure.



In addition to the above datasets describing the state of the atmosphere, we analyse ocean surface and subsurface temperature anomalies using ocean potential temperature data (THETAO) from the ORA-20C ocean reanalysis for the 20$^{th}$ Century (de Boisseson et al., 2018). The ORA-20C set was developed with the objective of providing ocean initial conditions for the coupled CERA-20C described above and is therefore consistent with it in terms of external forcings. It has 10 ensemble members covering the 1900-2009 period.

For validation purposes the following observational datasets are also examined: the NOAA ERSSTv5 sea surface temperature (SST) for 1854-2014 (Huang et al., 2017), the Met Office Hadley Centre / Climatic Research Unit HadSLP2r mean sea level pressure (SLP) for 1850-2014 (Allan and Ansell, 2006) and the HadCRUT5 near-surface temperature (TAS) dataset for 1850-2014 (Morice et al., 2021).





| | | DATASETS | | |
|---|---|---|---|---|
| TYPE | NAME | DESCRIPTION | PERIOD | SOURCE |
| OBS | ERSSTv5 | NOAA ERSSTv5 sea surface temperature (SST) (Huang et al., 2017). | 1854-2014 | https://psl.noaa.gov/data/gridded/data.noaa.ersst.v5.html |
| OBS | HadSLP2r | Met Office Hadley Centre/Climatic Research Unit HadSLP2r mean sea level pressure (SLP) (Allan and Ansell, 2006). | 1850-2014 | https://www.metoffice.gov.uk/hadobs/hadslp2/ |
| OBS | HadCRUT5 | Met Office Hadley Centre/Climatic Research Unit HadCRUT5 near-surface temperature (TAS) dataset (Morice et al., 2021). | 1850-2014 | https://www.metoffice.gov.uk/hadobs/hadcrut5/ |
| REA | ERA5 | ECMWF Reanalysis v5 (Hersbach et al., 2020). Variables: TAS, SLP, zonal wind at pressure levels (UA), air temperature at pressure levels (TA). | 1950-2014 | https://cds.climate.copernicus.eu/#!/search?text=ERA5&type=dataset |
| REA | CERA-20C | ECMWF 10-member ensemble of coupled climate reanalyses of the 20th century (Laloyaux et al., 2018). Variables: TAS, SLP, UA, TA. | 1901-2010 | https://apps.ecmwf.int/datasets/data/cera20c |
| REA | ORA-20C | ECMWF 10-member ensemble of ocean reanalyses covering the 20th century using atmospheric forcing from ERA-20C (de Boisseson and Alonso-Balmaseda, 2016). Variables: sub-surface ocean potential temperature (THETAO). | 1901-2009 | https://www.cen.uni-hamburg.de/en/icdc/data/ocean/easy-init-ocean/ecmwf-ensemble-of-ocean-reanalyses-of-the-20th-century-ora-20c.html |
| REA | 20CRv3 | NOAA-CIRES-DOE Twentieth Century Reanalysis v3 (Slivinski et al., 2019). Variables: TAS, SLP, UA, TA. | 1850-2014 | https://psl.noaa.gov/data/gridded/data.20thC_ReanV3.html |


**Table 1. Description of the datasets used in the study. Reconstructed observations are identified as OBS and reanalysis datasets are identified as REA.**



## 2.2 Methodology

The main tool employed to assess the 11-year solar cycle (SC) signal is a lead/lag multiple linear regression model (MLR) which has been widely used in the literature (Lean and Rind, 2008; Frame and Gray, 2010; Gray et al., 2013; Misios et al., 2016; Ma et al., 2018; Kuroda et al., 2022). We built an MLR model using multiple forcing indices (predictors) to describe the solar activity as well as additional sources of climate variability in the North Atlantic region. For the reanalysis datasets the 11-yr solar cycle is characterised by a detrended TSI predictor (DTSI hereafter) using a band-pass Butterworth filter to retain

periodicity between 7-15 years, as described in Misios et al. (2016). For the long reconstructed observations the solar sunspot number (SSN) is used instead, for consistency with previous analyses (e.g. Gray et al., 2013), but we note that the resulting 11-year SC signals are not sensitive to the choice of the solar predictor.

In addition to the 11-year solar cycle, a set of predictors that capture well-known sources of climate variability in the North Atlantic region is also included in the MLR. The choice of these predictors follows previous MLR studies (e.g. Frame and

Gray, 2010, Gray et al., 2013, Misios et al., 2016, Ma et al., 2018) but has also been the subject of thorough testing to avoid redundant predictors and co-linearity effects.

The globally-averaged stratospheric aerosol optical depth (AOD) at 550 nm (Sato et al., 1993) was used to represent major volcanic eruptions. A long-term trend index to characterise global warming was based on the $CO_2$ equivalent concentrations of all GHGs, as in Misios et al. (2016). The El Nino-Southern Oscillation (ENSO) was represented using the SST-based EN3.4

index (Trenberth, 1997) which was calculated for each dataset individually. Similarly, a predictor was derived from each dataset to represent the impact of Atlantic Multidecadal Variability through an SST-based AMO index, as defined in Trenberth and Shea (2006). The quasi-biennial oscillation (QBO) is known to have significant surface impacts (e.g. García-Franco et al., 2022), but observations of the QBO do not extend sufficiently back in time to provide an index to analyse the longer historical periods back to 1850. In this case, the QBO forcing index was excluded, following the approach of Gray et al. (2016), whilst

ensuring that the analysis period spanned exact multiples of the QBO so that potential end-effects were avoided.

The statistical significance of the resulting regression coefficients was assessed using a 2-tailed Student t-test where the p-values were subsequently corrected for false discovery rate using the Benjamini-Hochberg method (Benjamini and Hochberg, 1995). In the analysis of the 10-member ensemble there is potential for over-estimating statistical confidence if they are treated as independent, since the ensemble-members are all constrained by the same assimilated data. To avoid this, each ensemble

member was analysed separately, and the results are presented by showing the mean response with significance levels overlaid only where at least 70% of the members meet the p-value level, for consistency with typical ensemble assessments. Results from the individual analyses are provided as supplementary figures, where appropriate.





To assess the overall NH winter response, the MLR analysis employed December-February (DJF) averages, both for the forcing indices and the variable under study. In some cases, individual months were also considered. In each case, the longest possible common data period was employed to optimize the signal detection.

Wavelet analysis (Torrence and Compo, 1998) is also employed to analyse the spectral properties of individual time series as well as the co-variability between them using cross wavelet spectrum (Grinsted et al., 2004). Such methodologies have been recently applied to the detection of SC influences on climate (Narasimha and Bhattacharyya, 2010; Gruzdev and Bezverkhnii, 2020; Song et al., 2022). They are advantageous for not assuming a linear relationship and thus they also detect non-stationary responses. The methodology applied here follows the approach of Dimdore-Miles et al. (2021), by employing a Morlet wavelet and assessing statistical significance of the signals with respect to a theoretical red noise AR1 process.

## 3. Results

### 3.1 Winter surface impacts using multi-linear regression

Surface impacts of the SC on North Atlantic climate have been previously documented in both SLP and temperature (Roy and Haigh, 2010, Gray et al., 2013, Gray et al., 2016). Figure 1 presents results from the lagged regression analysis for DJF for the period 1854-2014 between SSN and HadSLP2r SLP (top), HadCRUT5 near-surface temperature (TAS) and ERSSTv5 (SST). Positive lags indicate that the solar forcing leads the surface response.



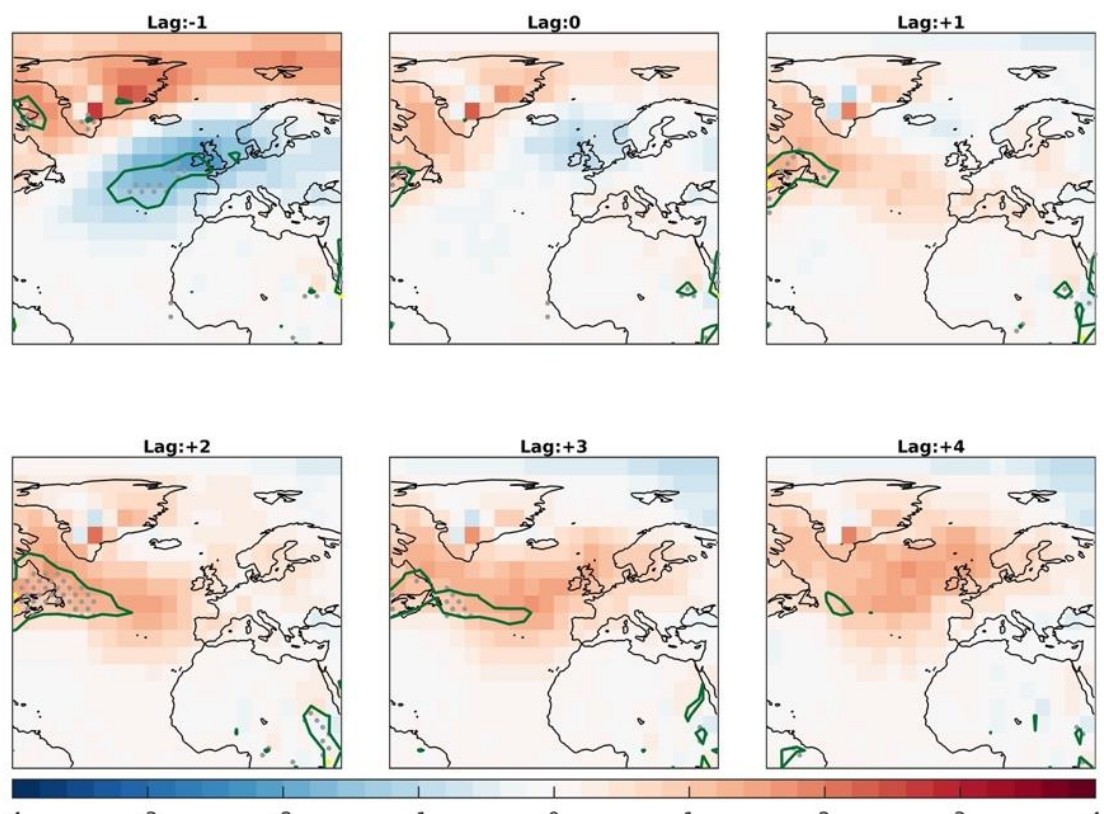



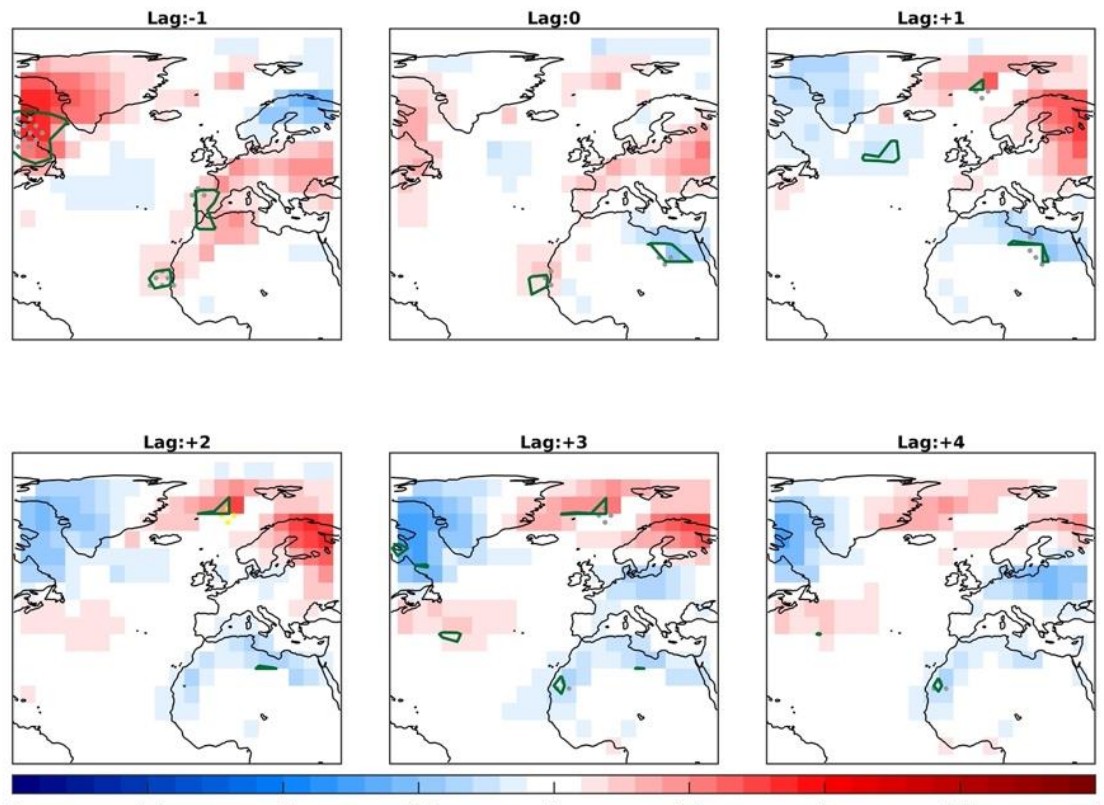





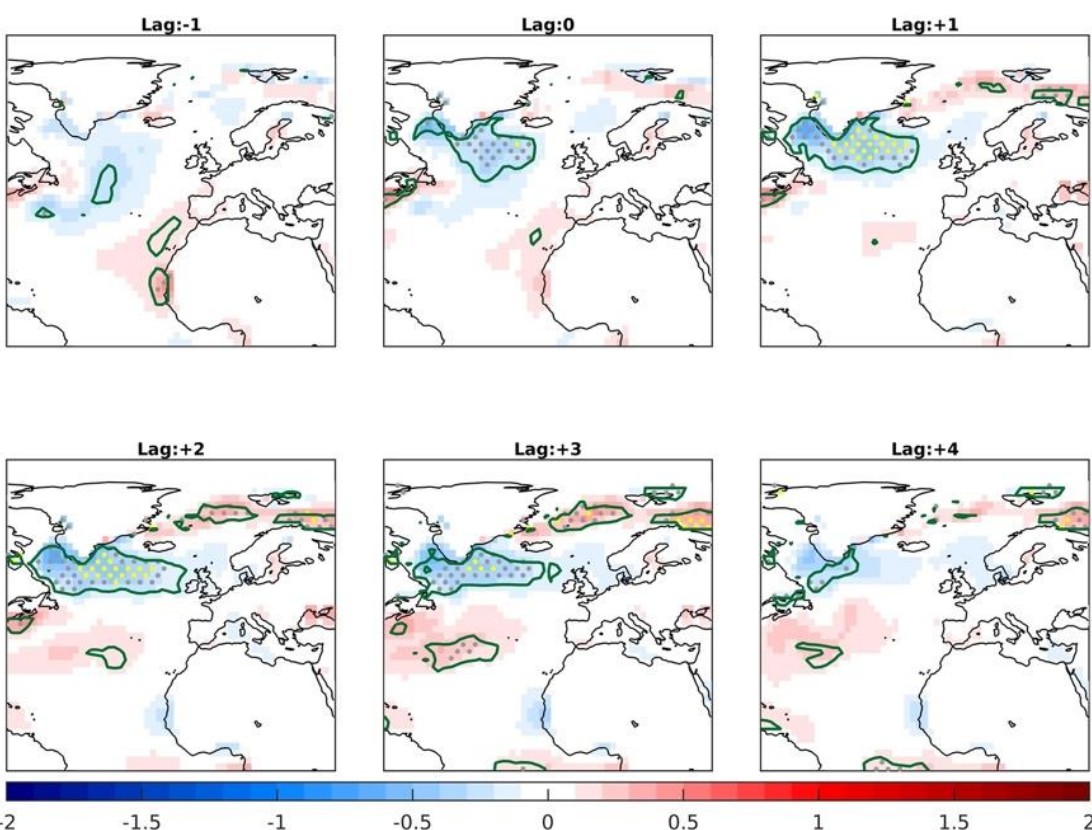


**Figure 1. Lagged solar regression analysis of reconstructed observational records for DJF over the period 1854-2014. a) HadSLP2r mean sea level pressure (SLP) in hPa; b) HadCRUT5 near-surface temperature in ˚C; and c) ERSSTv5 SST in ˚ C. Regression lags are expressed in years. Statistical significance is indicated by the green contour (90%), and the grey (95%) and yellow (99%) dots.**

In the case of SLP (Fig. 1a), the results are in reasonable agreement with previous studies (Gray et al., 2013; Gray et al., 2016;

Kuroda et al., 2022). They show a lagged response to the SC with a significant positive response in the region of the Azores,

peaking around lags of +3 yrs and a corresponding negative response approximately half a cycle earlier, the remnants of which

can be seen at lags of -1 yr. Brugnara et al. (2013) have identified qualitatively similar late-winter SLP anomalies at lag-0 in

gridded reconstructions, indicative of a more positive NAO index but they noted a low statistical significance. We note here

that the maximum positive response is not centred directly over the Azores but is slightly to the north, in good agreement with

the results of Gray et al. 2016; see their figure 3 at lag +3 yrs). As in previous studies, the responses further north over the

Icelandic region are not significant.

The corresponding TAS analysis using HadCRUT5 (Fig. 1b) shows the development of a quadrupole response structure

maximising at lag +3 yrs, with warm anomalies over northern Europe and the SE coast of the U.S. and cold anomalies over





the Labrador Sea and southern Europe, although there is no statistical significance. The SST analysis using ERSSTv5 (Fig.

1c) shows a clearer signal, with a statistically significant cooling response over the Labrador Sea and sub-polar gyre (SPG) region that peaks at lag +3 yrs and forms part of a tripolar pattern with warming responses on the southern and NE flanks.

Figure 2 presents the lagged solar regression responses for SLP using the three reanalysis datasets. The ERA5 reanalysis is only available for the more recent period 1950-2014 (the end date was chosen to coincide with the available end date of the other datasets). In this case the lagged North Atlantic response is spatially coherent and peaks at lag +3 yrs but is not statistically

significant. The 20CRv3 SLP solar response over the longer 1850-2014 period (Fig. 2b) shows an evolving response consistent with those in ERA5 but statistical significance is achieved and peaks at lag +3 yrs north of the Azores region. Some significant negative responses are also observed north of Iceland and in polar regions at this lag. Finally, the CERA-20C SLP solar response (Fig. 2c), which covers the period 1901-2010 and has 10 ensemble members, is consistent with the other two datasets.

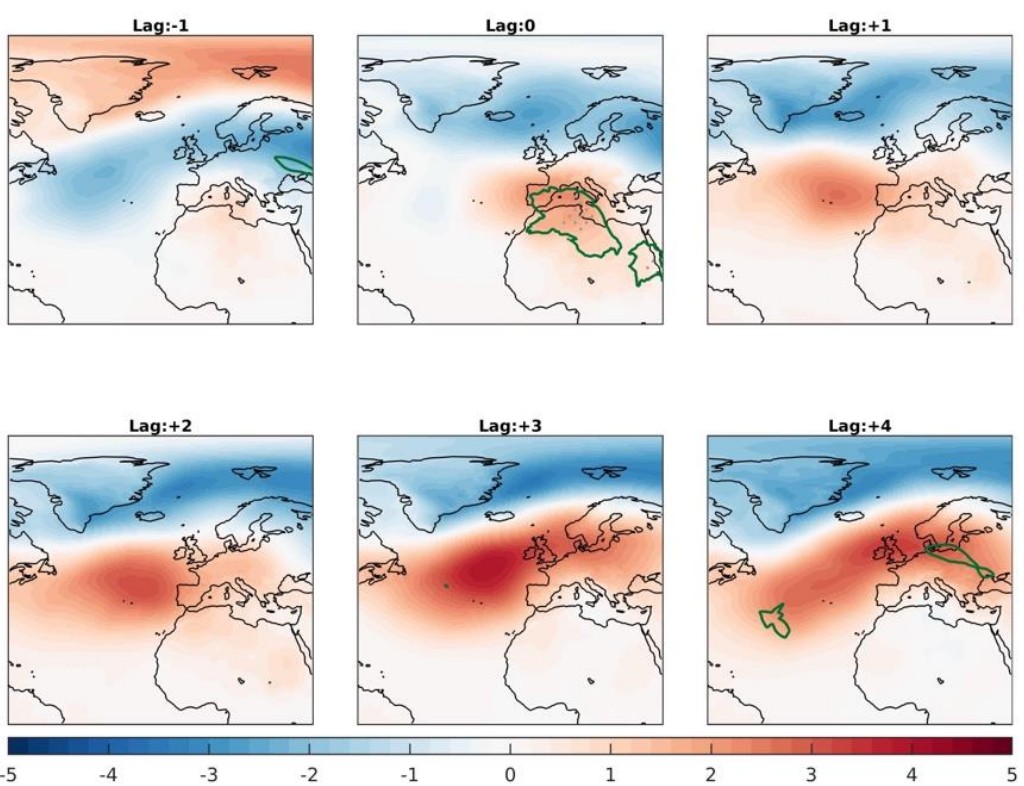




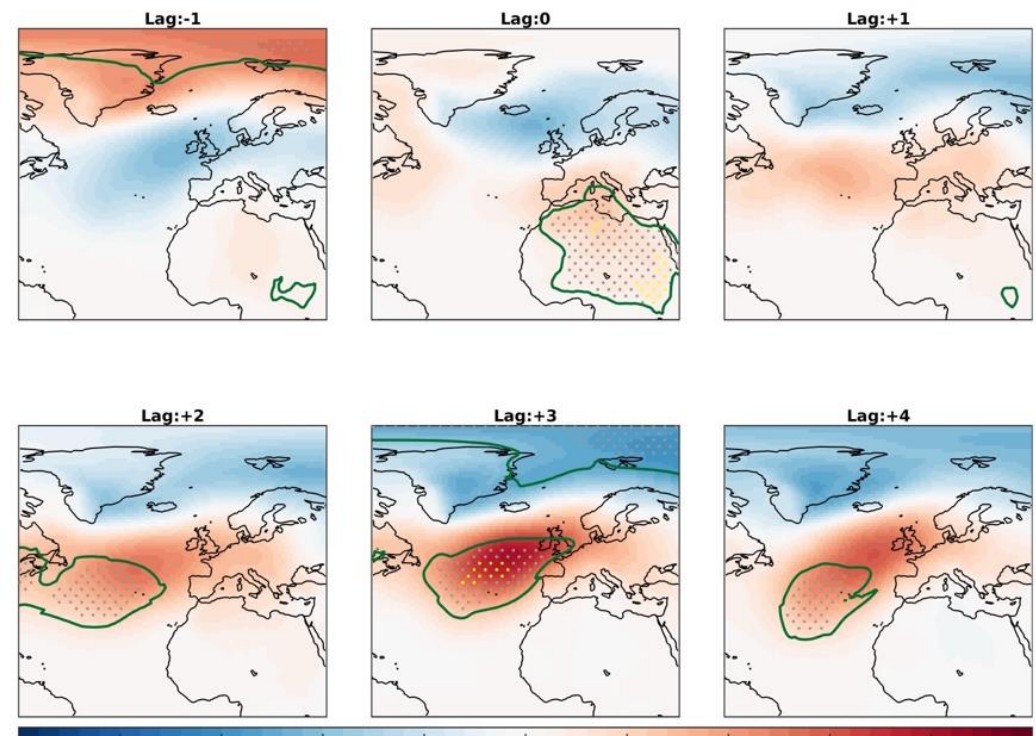



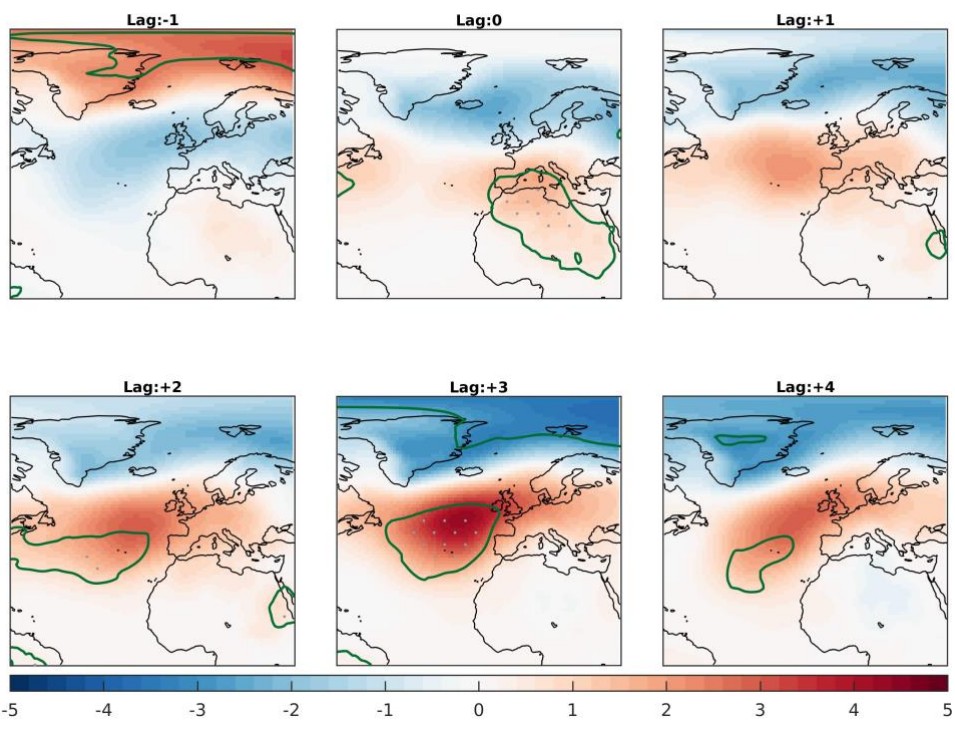

**Figure 2. Lagged solar regression analysis of DJF mean sea level pressure (hPa) using the reanalyses datasets. a) ERA5 1950-2014, statistical significance as in Fig. 1; b) 20CRv3 1854-2014, statistical significance as in Figure 1; c) CERA-20C 1901-2010 using 10 ensemble-members. In (c) the shading shows the ensemble-mean regression coefficients; levels of statistical significance are defined as in previous panels but only shown for grid points where the criteria is met by at least 7 of the 10 ensemble-members. Note the different color scales compared to Fig. 1a.**

Results of the corresponding analysis of DJF near-surface temperature (TAS/T2M) are presented in Fig. 3. In the case of ERA5

205 (Fig. 3a), the cold signal over the SPG region is significant at lag +1 yrs and by lag +3 yrs the quadrupole structure is well

established. In the case of 20CRv3 (Fig. 3b), the same evolution is observed, with high statistical significance particularly over

the NE U.S.A. and Labrador Sea regions. This structure is confirmed in the CERA-20C (Fig. 3c), with a near-identical

structure and evolution.



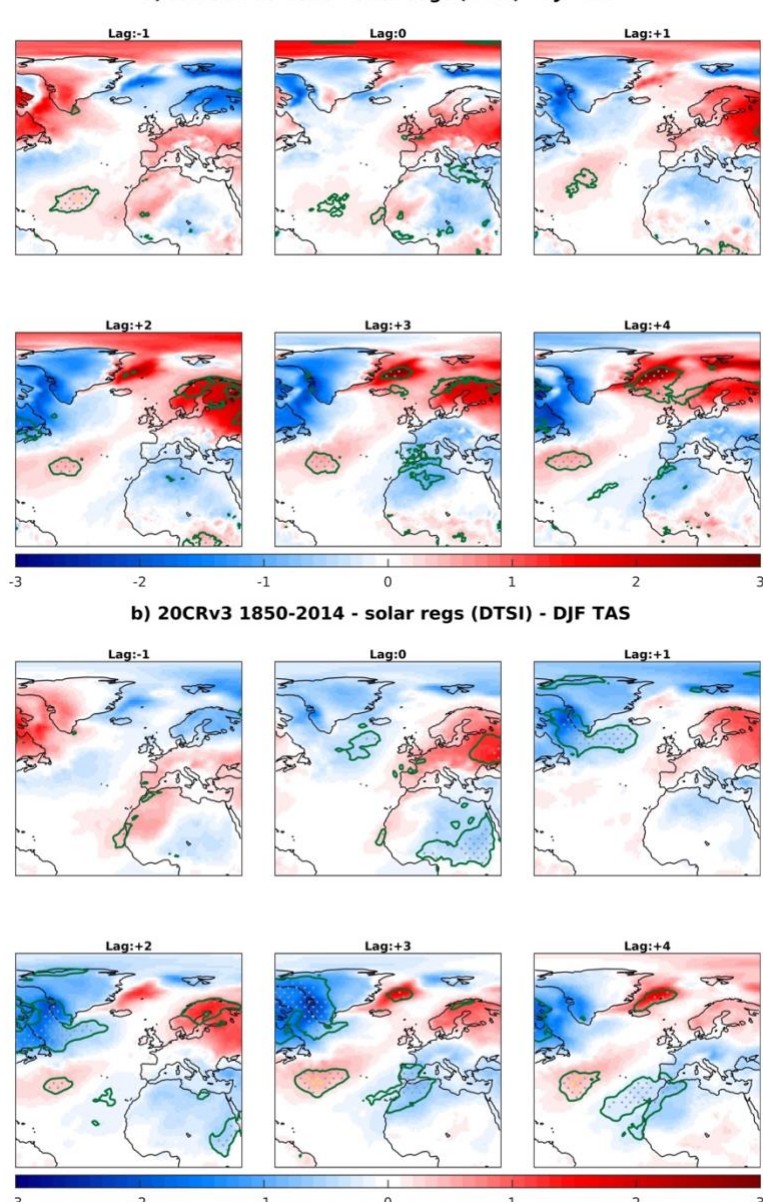

210



215

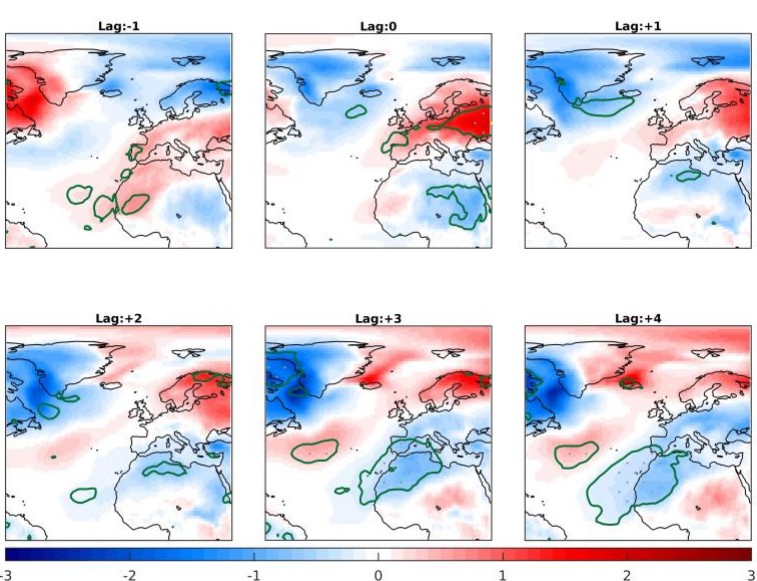

**Figure 3. As Fig. 2 but for DJF near-surface temperatures in ° C. Note the different color scales compared to Fig. 1.**

To explore the benefits of a 10-member ensemble, the 11-year solar response in DJF SLP from each individual CERA-20C ensemble-member at lag +3 yrs is shown in supplementary Fig. 1. They can therefore be compared with the 10-member averaged response shown in Fig. 2c. The individual ensemble members show that the response in the region north of the Azores is statistically significant at the 99% level for every individual ensemble member. The consistency in response between the ensemble-members is perhaps not surprising, since SLP is assimilated by the reanalysis process. Nevertheless, it demonstrates that the spread of uncertainty in SLP is relatively small, and this translates to increased confidence in the solar response. A similar analysis of the near-surface temperature (TAS) response in the individual ensemble members of CERA-20C (Supplementary Fig. 2) also reveals very good agreement between the single members and the 10-member average.

### 3.2 Winter surface impacts using sliding regression analysis

To explore the transient features of the solar response signals, a sliding regression approach was then implemented, following Ma et al. (2018). An area-averaged SLP index was selected over the Azores (30-15°W; 30-40°N) using the same latitude / longitude definition as employed by Ma et al. (2018) which encompasses the maximum response slightly to the north of the Azores in Fig. 2. The results were not sensitive to small changes in definition of the selected region. A lagged regression analysis was performed, for lags between -5-yrs and + 5-yrs, using a 45-year sliding window so that the time-evolution of the response could also be examined. Results from this sliding regression analysis of the Azores index are presented in Fig. 4.



Using the HadSLP2r observational dataset, Fig. 4a reveals that the lagged positive response at around lag +3 yrs is only significant for the most recent period, for windows centred from around 1960. For the earlier periods, a positive 11-yr solar response at positive lags is observed but with no significance and only significant negative responses at lags 0 and -1 yr are evident. The analogous results for the 20CRv3-based Azores index (Fig. 4b) show a similar result. Figure 4c shows the corresponding analysis from the first ensemble member of CERA-20C (which covers a shorter period and can be compared to the responses to the right of the vertical black lines in Figs. 4a,b). The comparison shows a high level of agreement with the HadSLP2r and 20CRv3 results and this is also true for all other CERA-20C ensemble members (not shown). Results from the corresponding analysis of the ERA5 data set are not shown here, since it covers a much shorter period and thus a 45-yr sliding regression analysis is less useful, but the spatial distribution of the results are very similar to those in CERA-20C.





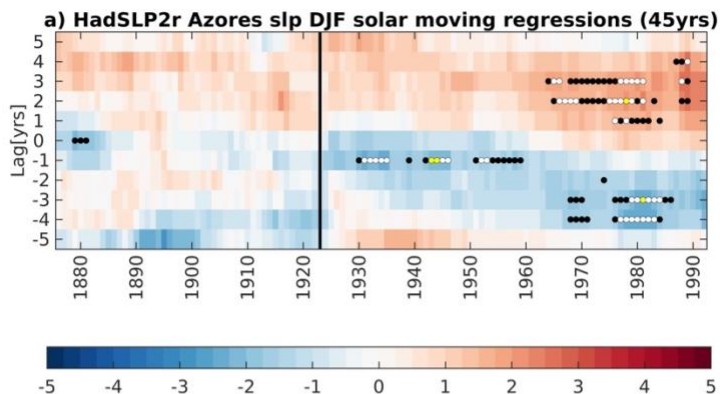

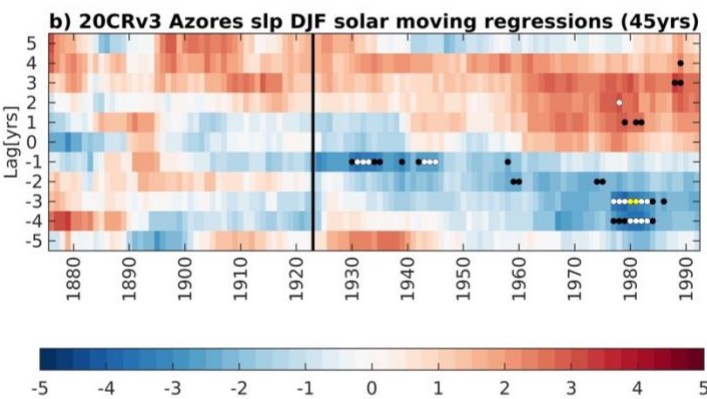

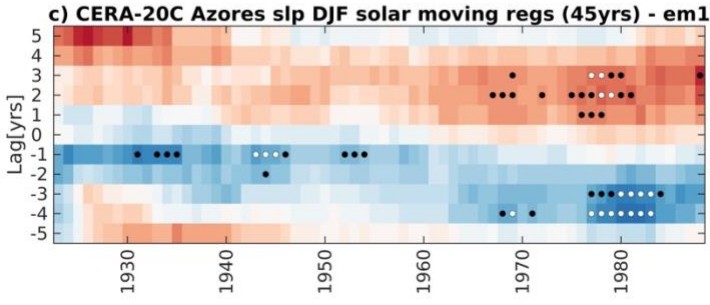



**Figure 4. Solar response of DJF Azores SLP (hPa) from multiple linear regression analysis using a moving 45-year window: a) HadSLP2r 1854-2014; b) 20CRv3 1854-2014; and c) CERA-20C 1901-2010 (member 1). X-axis labels identify the central year in the 45-year window. The vertical black line in panels (a) and (b) indicates the start of the period covered by CERA-20C in panel (c). Statistical significance is indicated by the black (90%), white (95%), and yellow (99%) circles.**

These sliding regression results are consistent with those presented by Ma et al. (2018) in which only the HadSLP dataset was examined. Ma et al. (2018) proposed that the non-stationarity in the SC response could partly be due to variations in the amplitude of the solar forcing, which has varied over time (see Ma et al. 2018 for further discussion). The results in Fig. 4 also suggest a slight shift of the response between the mid-century period (1930-40) where the main response occurs at lags of +4-5 yrs and the later part of the century where the response occurs at lag+3 yrs. In all three datasets the response at lag-zero is small and statistically insignificant at all times. Because of this, it is easy to see why previous studies that have examined only the lag-zero response failed to find any level of statistical significance (e.g. Chiodo et al., 2019). Nevertheless, these results also show that the 11-year solar response over the Azores is relatively weak over most of the past century, the response is non-stationary and dominated by the period since ~1960 when improved observations became available.

Figure 5 shows results of the sliding regression analysis for the Azores index calculated for individual months in the December-February period. As before, the first ensemble member of CERA-20C is included in the right column and is representative of all ensemble members. In agreement with Gray et al. (2016) and Ma et al. (2018), the responses in all three reanalysis datasets show that in the early winter months (Dec-Jan) there are positive lagged responses over the Azores region at lags of +3 yrs and greater. In late winter (Feb) there are responses of the opposite sign in the earlier periods (pre-1920) of both HadSLP2r (left column) and 20CRv3 (central column), with varying levels of significance, but the more recent period (1960 onwards) shows significant positive responses at slightly smaller lags (+1-3 yrs) than in the early-winter months.



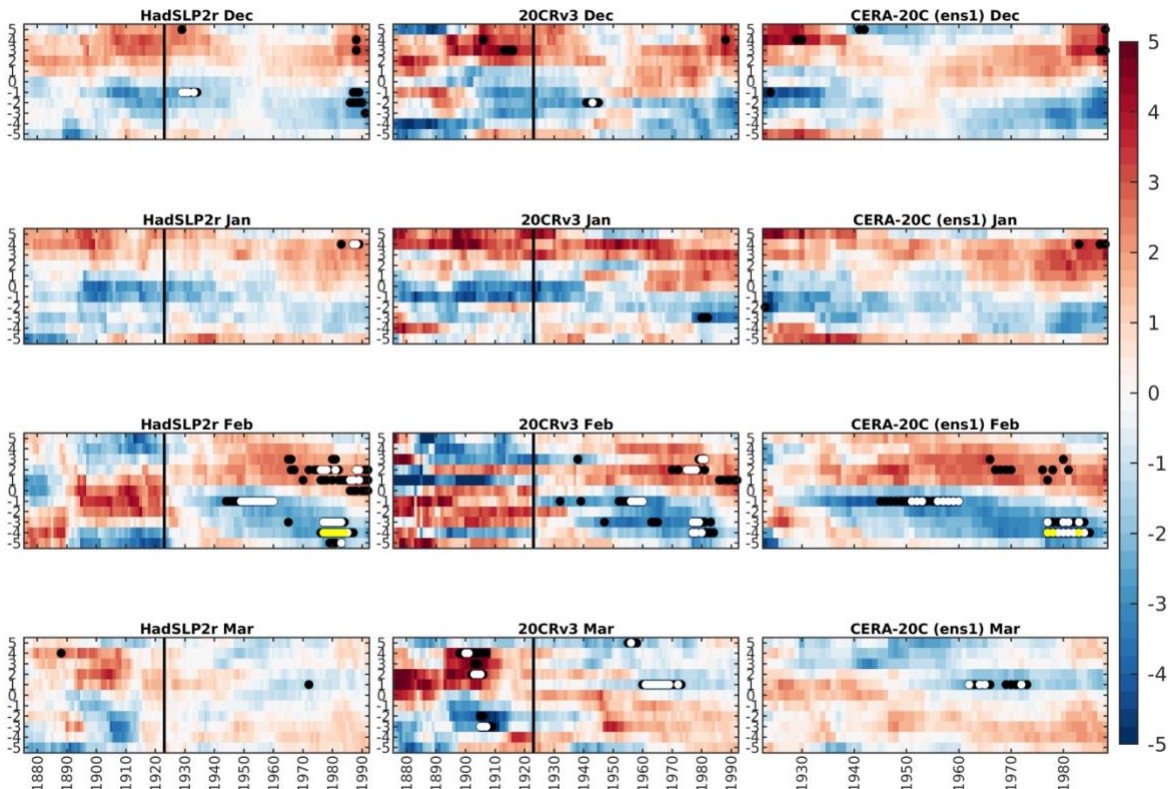

**Figure 5. Solar response of monthly-averaged SLP (hPa) over the Azores region from a multiple linear regression analysis using a moving 45-year window in HadSLP2r (left column), 20CRv3 (central column), and CERA-20C ens1 (right column). The vertical black line in the two leftmost columns indicates the start of the period covered by CERA-20C in the right column. Statistical significance as in Figure 4.**

### 3.3 Winter surface impacts using wavelet analysis

The non-stationarity of the observed relationship in Fig. 4 between the 11-yr solar forcing and surface impacts suggests that a linear approach, such as the widely-used multi-linear regression analysis, is likely to be sub-optimal for extracting information about such influences. As an alternative and complementary methodology, we now examine the spectral properties of the individual timeseries, as well as their co-variability using wavelet analysis. As noted in Section 2, the main benefit of the wavelet analysis approach is that it can more accurately identify non-stationary behaviour. Fig. 6 (a-c) shows the wavelet power spectra for the DJF Azores SLP index from the three reanalysis datasets. As a reference point, the wavelet power spectrum of the solar indices SSN and DTSI are also shown (Fig. 6d and 6e respectively). The latter both show a statistically significant peak in the 8-16 years period band, confirming the presence of the 11-yr solar signal, with stronger spectral



densities (meaning stronger variability) towards the end of the period. The SSN spectrum additionally shows power at longer

periods due to the presence of long-term solar variability (which has been removed from the DTSI index – see section 2).

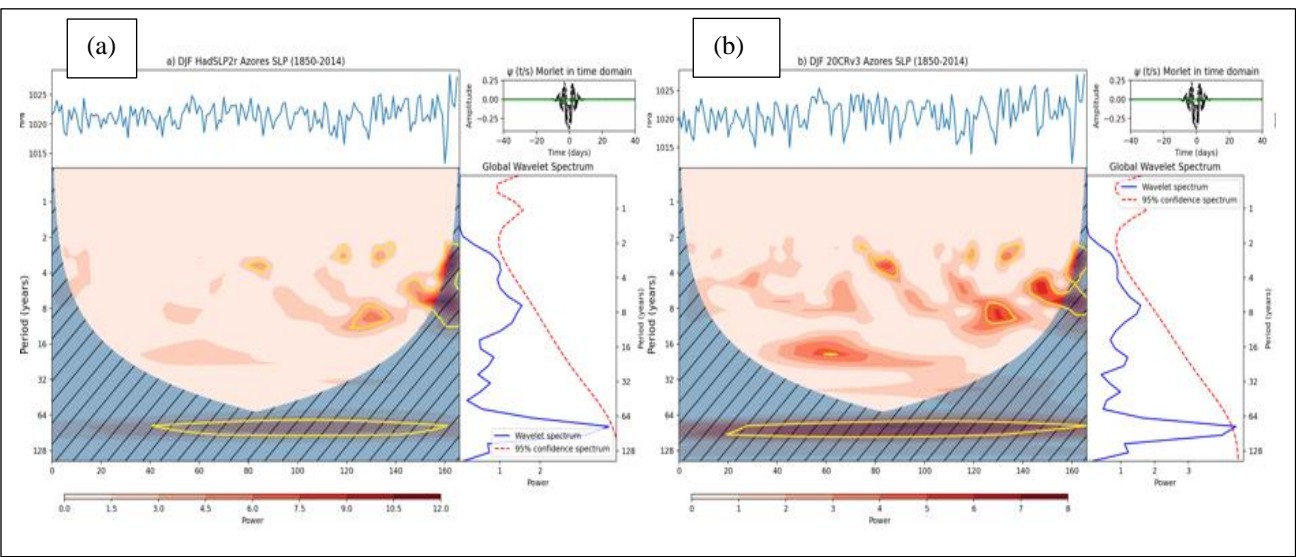

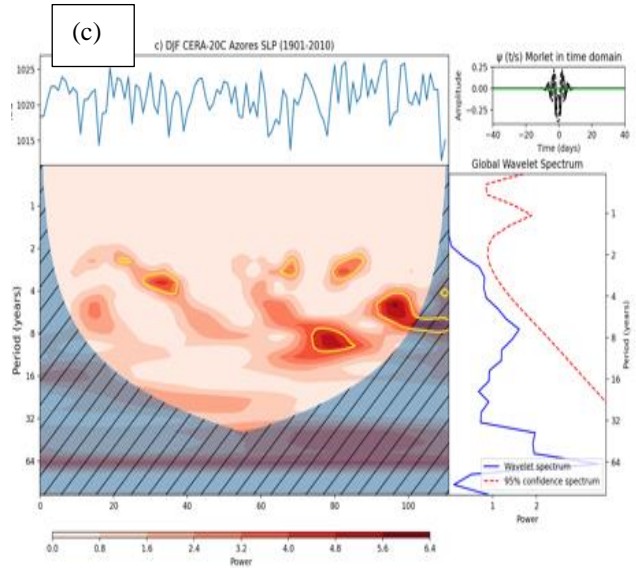





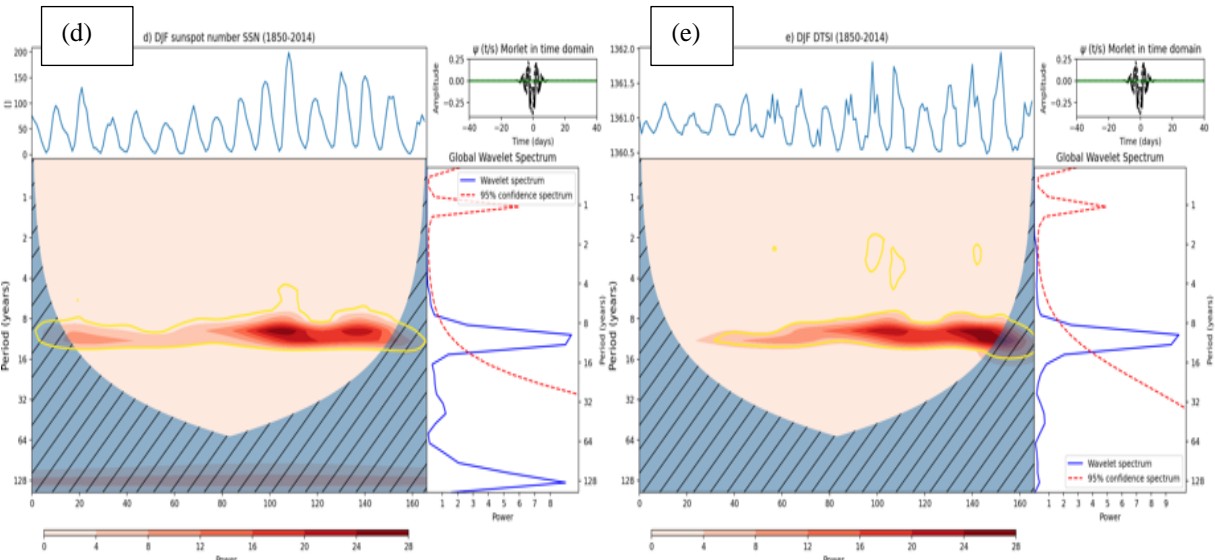


**Figure 6. (a-c) Wavelet power spectrum of the DJF Azores SLP timeseries from the following datasets: a) HadSLP2r, b) 20CRv3, c) CERA-20C SLP (member1). For comparison the corresponding power spectra are shown for d) the SSN solar index, and e) the DTSI solar index. Statistical significance with respect to a background AR1 red noise is indicated by the yellow contour. Hatching indicates area where edge effects are important. In each panel, the top right diagram presents the Morlet wavelet used in the transform while the bottom right plot presents the global power spectra (blue curve) and the corresponding global 95% confidence levels (red curve).**


Even though the spectral densities of the HadSLP2r, 20CRv3 and first ensemble-member of the CERA-20C dataset (Fig. 6a-c) show statistical significance in the 8-16 yr band for some time periods (shown by the yellow contour), the corresponding

peak in the global wavelet spectrum (on the right-hand side of each plot) is not statistically significant. However, when all 10 ensemble members of the CERA-20C dataset are combined as a single time-series, the peak in the global wavelet spectrum at 8-16 yrs extends above the red noise background spectrum and is thus globally significant (see Supplementary Fig. 3).

To further explore the co-variability between the 11-yr solar forcing and the SLP Azores indices, a cross-wavelet power spectrum analysis was performed. Figure 7 shows results of the cross-wavelet power spectrum analysis between the DJF-

averaged DTSI/SSN solar forcing index and the 3-yr lagged Azores SLP index from the HadSLP2r dataset, the 20CRv3 dataset and the 1[st] ensemble-member of the CERA-20C dataset. The solar index selected for the analysis depended on the length of the dataset under examination; where possible, sensitivity tests were performed by repeating the analysis with both DTSI and SSN indices to check that there were no major differences in the results. The time-series of the two indices are shown at the top of each plot. The main panel shows their joint spectral density (colour shading) and a measure of their phase

agreement (arrows). Arrows pointing towards the right (left) indicate an in-phase (out-of-phase) relationship between the two indices, while directions between these extremes indicate an intermediate phase relationship e.g. where one of the signal leads or lags by $\pi/2$. Results from all three reanalysis datasets are clearly consistent and show joint spectral densities



in the 8-16 yr period band that are statistically significant. This is also true of all ensemble-members of CERA-20C as shown in Supplementary Fig. 5. These strong relationships coincide with the epoch of strong solar variability in Fig. 6d and 6e. Over

the areas of strong, significant joint density, the arrows are mostly pointing right, which indicates that the indices are in phase with each other. Note that the Azores index for the reanalysis datasets are defined at a lag of +3 yrs, so this simply means that the solar forcing leads the Azores SLP response by ~3 yrs.









**Figure 7. Cross wavelet power spectra between pairs of DJF timeseries: a) SSN vs. HadSLP2r Azores SLP index, b) DTSI vs. 20CRv3 Azores SLP index, and c) DTSI vs. 1$^{st}$ ensemble member of the CERA-20C Azores SLP index. The Azores indices lag the solar index by +3 yrs. Statistical significance with respect to an AR1 red noise background is indicated by the yellow contour. Arrows pointing towards the right (left) indicate an in-phase (out-of-phase) relationship between the two indices.**

In summary, the wavelet analysis of the 11-yr solar cycle indices and the Azores SLP indices is independent of the MLR approach. The results reinforce the MLR results by showing a statistically significant impact of the 11-yr solar cycle on the SLP in the region of the Azores at lags of +3 yrs. The analysis additionally shows that the relationship is particularly robust for the epoch when the solar forcing was strongest, starting from around the mid-1940s.

**3.4 The role of the ocean in the lagged response**

Previous work has suggested that the ocean may play an important role in the delayed response of the North Atlantic SLP to the solar cycle due to its thermal inertia. In particular, a re-emergence of subsurface thermal anomalies in the years following solar maxima has been proposed as a mechanism to explain the 2-3 year lagged response (Scaife et al., 2013; Andrews et al., 2015). Exploiting the fact that the CERA-20C dataset is a coupled retrospective reanalysis, this section focuses on the signals of the SC on the subsurface temperatures of the companion ORA-20C dataset, which was used to provide initial conditions for the CERA-20C reanalysis.

To determine the relevant oceanic regions on which to focus, the relationship between various surface variables over the North Atlantic in CERA-20C was examined. Figure 8 presents the correlations between the CERA-20C SLP Azores index and the surface temperature field for DJF. In contrast to Andrews et al. (2015), the focus here is on the Azores region rather than the NAO index, given that this region showed the most robust SC signal. Nonetheless, the tripolar pattern in Fig. 8 is very similar to the NAO temperature response pattern found by Andrews et al. (2015) and many other studies of the NAO. Figure 8 can be used to identify two regions with the highest correlation values that represent the northern (N-trip) and central (C-trip) regions of the Atlantic tri-pole. The southernmost tri-pole region shows a weaker relationship to the Azores SLP and is therefore omitted from the subsequent analysis. Additionally, we also focus on a region in the East Pacific (Epac) since SSTs over this region have previously been shown to exhibit a SC response (Meehl et al., 2009; Tung and Zhou, 2010; Lin et al., 2021).



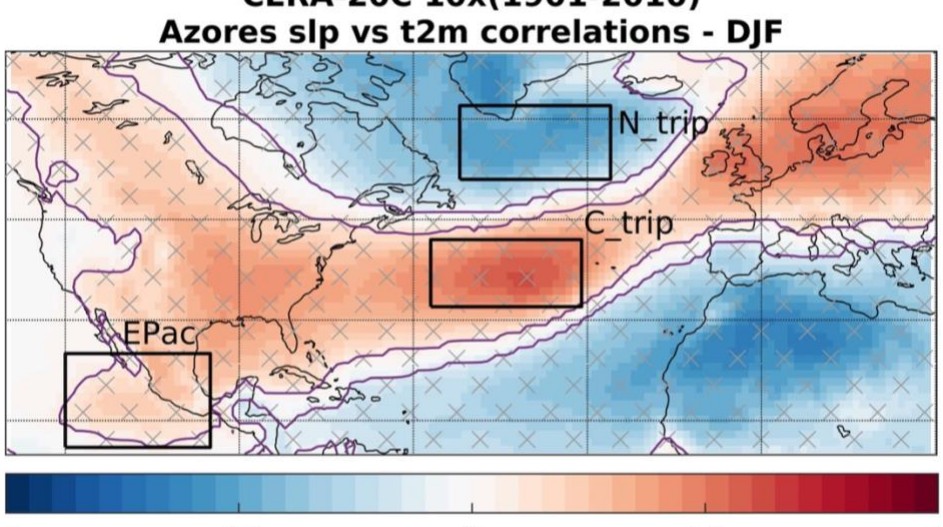

**Figure 8. Correlation coefficients between the Azores DJF SLP index and near-surface DJF temperatures in CERA-20C. Statistical significance at the 99% levels is indicated by the purple contour and at the 90% by the grey hatching. The correlations were performed for 1901-2010 using 10 ensemble members.**

Figure 9 presents the SC signal from the MLR regression analysis of ORA-20C ocean potential temperatures (THETAO) for the regions identified above (using the monthly DTSI as the 11-yr SC forcing index). The regions show a clear and statistically significant signature of the 11-yr SC, modulated by the seasonal cycle. The responses vary not only in sign and strength but also in the depth to which they extend, with maximum responses that vary from close to the surface down to more than 150m in depth. This variation in depth extension is even more clear when exploring the regressions all the way through 600m depths (Supplementary Fig. 6). The shallowest responses are observed for the Epac box (see Supplementary Fig. 6c) as expected for a region associated with eastern boundary upwelling. In contrast, the deepest impacts are seen for the N-trip box (Fig. 9a and Supplementary Fig. 6a), which is also expected for a region with significant deep-water formation. This northernmost box experiences a significant cooling signal following solar maxima that propagates downwards and extends to depths below 200m. The C-trip box (panel b) shows a warming response that starts after the solar maxima and also propagates downwards.

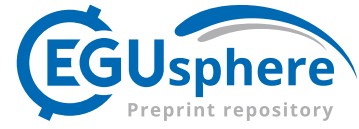

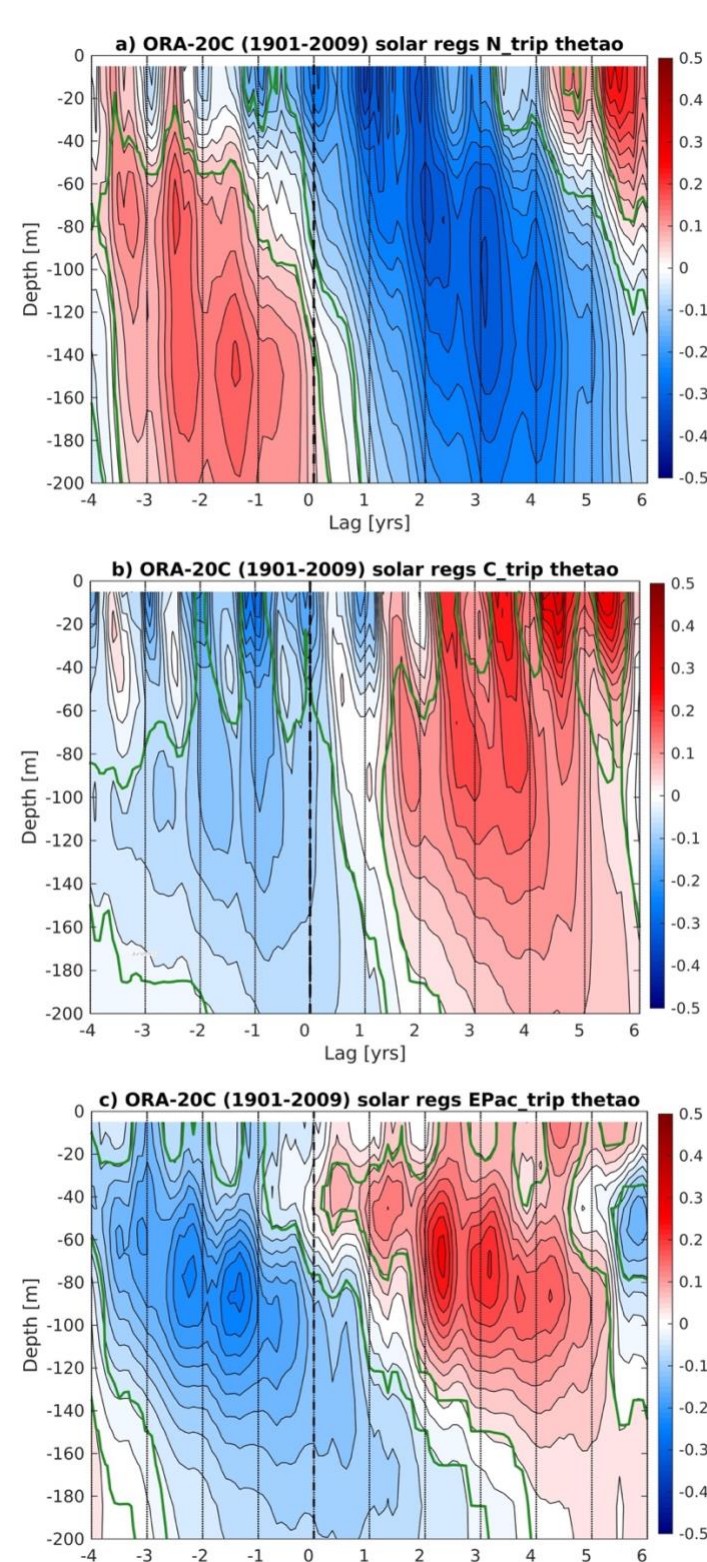





**Figure 9. Solar response from multiple linear regression analysis of monthly subsurface THETAO (ºC) between 0-200m ocean depth for the following boxes indicated in Fig. 8: a) N-trip ; b) C-trip; and c) EPac. Statistical significance at the 95% levels is indicated by the green contour. The regressions were built for 1901-2009 using 10 ensemble members.**

More interestingly, the regions identified above exhibit signs of re-emergence of the sub-surface warming in subsequent

years, reinforcing the surface warming. This results in a signal that peaks near the surface between lags of +3 and +5 years, particularly evident in the C-trip region centred close to the Azores. These results are consistent with the model-based results of Andrews et al. (2015). This reinforced near-surface warming is linked to positive SLP anomalies over the Azores region, consistent with an NAO+ pattern. The Epac box (Fig. 9c) also shows a signal of warming following SC maxima but its downwards propagation is shallower and appears slower. A marked difference in this region is the fact that the warming

signal does not peak near the surface, but rather at depths of between 30m and 90m. This is likely due to the differential ocean dynamics of this region compared to the North Atlantic focus boxes and could be associated with thermocline depth adjustments to surface wind responses (Misios et al., 2019). Nonetheless, the analysis also suggests the presence of a re-emergence signal after the solar maxima, with a relative maximum in the surface warming at lag +4 yrs.

To explore these connections in more detail, Figure 10 presents a 45-yr moving-window multiple linear regression analysis

between the DTSI index and THETAO at a depth of 10m over the three identified boxes. For all three regions a relatively stable pattern of the same sign is evident over the full ORA-20C period, although the response is only statistically significant over the most recent period (since 1950). The analysis shows regional ocean cooling (warming) over the N-trip (C-trip) region that peaks at lags of +3-5 years. As indicated by Fig. 8 these anomalies are consistent with positive SLP anomalies over the Azores region at similar lags, further supporting the proposed mechanism that thermal inertia of the ocean may be

responsible for the lagged response to the 11-yr SC in the North Atlantic SLP.



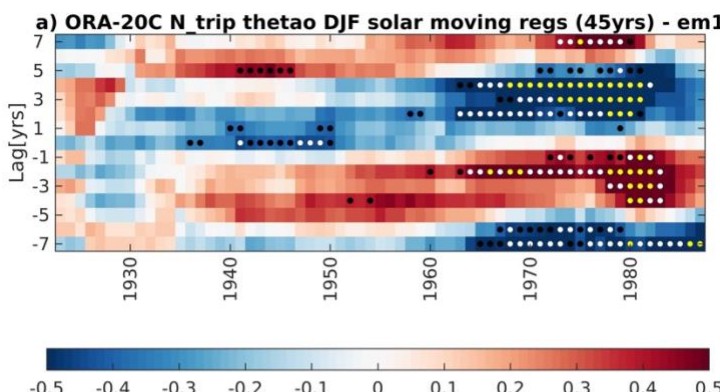

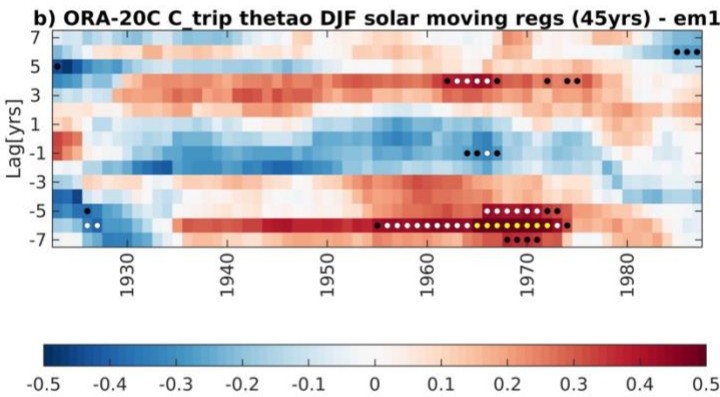

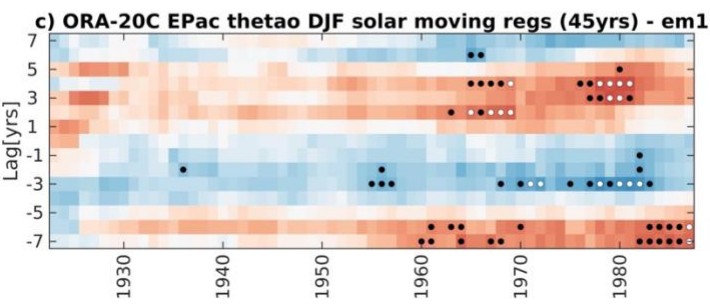



**Figure 10. Solar response in monthly-averaged ORA-20C THETAO ocean temperatures (ºC) at 10m depth from multiple linear regression analysis with a moving 45-yr window for the boxes identified in Fig. 8: a) N-trip ; b) C-trip; and c) EPac. X axis labels identify the central year in the 45-yr window. The regressions were performed for 1901-2009 and correspond to the first ensemble member. Statistical significance is indicated by the black (90%), white (95%), and yellow (99%) circles.**

## 4. Discussion and conclusions

This study has examined a set of reanalyses, both modern and 20th Century, to evaluate the robustness of the signatures of the 11-yr solar cycle in the North Atlantic climate. 20th Century reanalyses have the advantage of providing a longer representation of the climate than modern ones and, in some instances, they include multiple ensemble members. Additionally, when created as a coupled ocean-atmosphere reanalysis they have companion long-duration oceanic datasets that are physically consistent and allow further exploration of the presence of subsurface signatures of the SC.

The study employed two different statistical methods, one of which (MLR) assumes a linear relationship between the solar forcing and the impacts, and one of which (wavelet analysis) does not. Both techniques yielded consistent results. The use of such datasets and tools, and their comparison with long-reconstructed observational datasets are fundamental to demonstrate that the North Atlantic surface climate has a robust signature of the 11-yr SC. This solar cycle response is mainly characterized by positive anomalies in SLP in the region slightly to the north of the Azores at lags of ~3 yrs that contribute to a positive NAO-like pattern. This 11-yr SC signal is not, however, a stationary feature. The influence of the 11-yr solar cycle is shown to vary not only within the winter season but also on multi-decadal timescales. The multi-decadal variability is evident in both the amplitude of the signal and in the lag of the response. This could be linked to the intensity of the forcing itself, as suggested by Ma et al. (2018), since solar variability has been stronger over the more recent period, but this requires further investigation.

Results of the analyses performed in the study are summarized here:

- We find a robust response to the 11-yr solar cycle over the North Atlantic sector with a positive SLP anomaly north of the Azores region at lags of +2-3 years following solar maximum. This signature of the SC was found using a multiple linear regression approach and was consistent across all the datasets examined, which included both contemporary and 20th Century reanalyses as well as observational datasets. The results confirm several previous studies (Gray et al., 2013; Thieblemont et al., 2015; Drews et al., 2022; Kuroda et al., 2022).
- A sliding 45-yr window regression analysis of SLP over the 20th Century revealed that there are non-stationarities in season and epoch of the 11-yr SC impacts. The lagged positive SLP response over the Azores region is a persistent feature of the early winter, but significance is achieved only during the strong solar forcing epoch in the second half of the century.
- The SLP response over the Icelandic region is barely significant, changes signs along the historical record, and is only consistent with an NAO+ structure during the recent strong solar forcing epoch. This suggests that the relatively short length of the available datasets prevents the detection of a signal above the background internal variability in this region. Also, the position of the maximum SC response to the north of the Azores suggests that it may not be appropriate to interpret the response purely in terms of the NAO. Similarly, Brugnara et al. (2013), found





insignificant correlation between the solar activity and the NAO index after analysing a 240-yr long reconstruction. Moreover it was found that the correlation of between the solar activity and SLP over the North Atlantic characterized more the late winter months, but any time delay of the signal was not examined. EOF analyses indicate the major North Atlantic variability (EOF-1) is associated with the NAO pattern but there is substantial variability in the lower-order EOFs, such as the second mode of variability (EOF-2) known as the East Atlantic pattern (see e.g. Fig. 9 of Barnston and Livezey, 1987). Further exploration of the 11-yr SC association with these additional patterns and with different blocking indices would be useful.

- An independent analysis using wavelet cross-spectra confirms the presence of significant co-variability between the Azores SLP and the 11-yr SC, with some dependence on the strength of the solar forcing. The transient features of the solar-Azores co-variances are consistent with those obtained from the sliding 45-yr regression analysis, but without having to assume a linear relationship between them.

- An assessment of the impacts of the SC on sub-surface ocean temperatures supports the hypothesis that the thermal inertia of the ocean can explain the observed lagged surface response. In particular, the re-emergence of warm anomalies in subsequent years after a solar maximum is observed over the central North Atlantic, in a region where surface warming is linked to positive SLP responses over the Azores and an NAO+. A sliding 45-yr regression analysis between DTSI and the 10m THETAO ocean temperature dataset revealed responses that were stable in sign and peaked at lags that support the theory that oceanic thermal inertia can explain the lagged surface response to the SC.

- An ocean response to the 11-yr SC was also identified in the eastern Pacific, with evidence of re-emergence at lags of +4-5 years. Warm temperatures over this region are consistent with positive Azores SLP, possibly linked through a teleconnection process (e.g., via a Pacific North Atlantic wavetrain) but this requires further investigation. Previous studies have primarily focused on stratospheric 'top down' forcing of the North Atlantic SLP response. This result, however, raises an alternative possibility that the North Atlantic SLP lagged response to the 11-yr solar cycle could be at least partially explained by forcing via the Pacific Ocean.

**Acknowledgments**

The work of PLMG, LJG and SO was funded by the NERC North Atlantic Climate System Integrated Study (ACSIS; grant no. NE/N018028/1) and the National Centre for Atmospheric Science ODA national capability programme ACREW (grant no. NE/R000034/1), which is supported by NERC and the GCRF. SM acknowledges the Villum Experiment grant "Environmental Consequences of Solar Cosmic Rays"

The sources of all the data used in this study are listed in Table 1.

The multiple linear regression model was implemented using the R-package lm (https://stat.ethz.ch/R-manual/R-devel/library/stats/html/lm.html).

Wavelet analysis was implemented in python using sample scripts provided by Oscar Didmore-Miles (https://github.com/oscardm20994/Wavelet_analysis; Dimdore-Miles et al., 2021).

**Conflict of Interest**

No conflict of interest is declared.



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
