# Peer review of "Solar cycle impacts on North Atlantic climate"

_EGUsphere, 2024_

## Referee Comment (RC1)

**Solar cycle impacts on North Atlantic climate**
**Submitted by Gonzalez et al. to WCD**

Gonzalez et al. are attempting to shed light on a long-standing problem in natural climate variability: the potential imprints of the 11-year solar cycle on weather and climate dynamics in the North Atlantic sector. To this end, they analyze observational data and reanalysis using multiple linear regression (MLR) and wavelet analysis.

While I was initially excited to read the manuscript, I soon realized that it does not present a significant amount of new information. It appears to be a reproduction of studies that have been conducted many times before, with the exception of the subsurface ocean analysis. In its current form, I do not believe this manuscript meets the criteria for publication in Weather and Climate Dynamics, and I doubt that it will undergo meaningful improvement through a major review. Therefore, I suggest that this manuscript be rejected.

**Major points of critism**

**Cited literature and discussion**
Unfortunately, Gonzalez et al. primarily cite outdated literature or studies that support the existence of the 11-year solar cycle in surface data of the North Atlantic during winter (with only a brief mention of Chiodo et al., 2019). This approach may not fully reflect the latest advancements in the field, particularly in light of recent studies that highlight numerous challenges with the solar cycle signals. Particularly noteworthy is that the most recent research increasingly highlights numerous flaws and challenges associated with the so-called 11-year solar cycle signals, which are evident even from the stratosphere, the origin of these signals according to the top-down mechanism. **To name just a few:**

**Chiodo et al. 2019:** The key point of criticism is that the quasi-decadal (11-year) variations in the North Atlantic SLP (sea level pressure), often attributed to solar variability, may actually be manifestations of internal variability rather than external solar forcing. The results show that the NAO (North Atlantic Oscillation) can naturally fluctuate at decadal timescales due to ocean–atmosphere coupling, with solar cycle forcing having a weak impact compared to internal variability. The low signal-to-noise ratio suggests that including solar forcing does not significantly improve prediction skill, and previous studies may have overestimated the influence of solar variability.

**Spiegl et al. 2023:** The main criticism, based on a highly sophisticated decadal prediction system (MiKlip historical ensemble simulations), is that the observed decadal solar signals are likely the result of internal tropospheric variability rather than the top-down solar mechanism. While solar signals were detected in the upper stratosphere, they did not consistently reach the surface, and correlations with the NAO were weak and inconsistent. Additionally, extreme variability in the North Atlantic sector quickly wipes out these small signals. Earlier model studies, which did not account for the full spectrum of naturally induced variability, may have overestimated the strength of solar signals as a result.

**Whuo et al. 2024a:** The key criticism concerns the proposed solar-NAO connection. Previous studies have suggested links between solar irradiance and the NAO, but the results remain debated due to mixed findings. In this study, the solar-NAO connection is only evident in the ensemble mean, while it is absent in many individual runs. This indicates that internal

variability might be masking the solar signal in shorter simulations. The weak solar signals require ensemble simulations to separate them from natural variability, highlighting that a single simulation is insufficient to capture such subtle effects. Additionally, observational data represent only a single member, making it challenging to provide clear physical insights without the models.

**Who et al. 2024b:** Discrepancies between models and from one ensemble member to another complicate the identification of solar signals amid internal variability, and the limited observational period adds to the challenge. Significant solar signals are observed at the tropical tropopause, but dynamical responses differ considerably across models and ensemble members. This suggests that linear analysis methods may not adequately capture these dynamics. Additionally, surface signals are irregular and not consistently visible with each solar cycle, raising concerns about the robustness of the findings related to both surface responses and the NAO connection. Biases in model temperature and wind speeds impact solar responses, particularly regarding the polar vortex, which can dampen or distort the solar cycle's influence on the meridional temperature gradient and zonal wind responses, further questioning the overall robustness of the conclusions.

A study on potential solar cycle signals that seeks to be published in 2024 should thoroughly address these issues and evaluate them critically. Unfortunately, Gonzalez et al. opted for a different approach. They attempt to explain subsurface solar signals in the ocean without first fully understanding the stratospheric pathway, where many problems already arise. It feels as though we are trying to address step two before step one. This critical analysis is absent from the current manuscript, yet it is absolutely essential, and reanalysis does provide the necessary data.

**Data and Methods**
Gonzalez et al. claim to be the first to analyze different reanalysis products in relation to the solar cycle. However, the reason this has not been done previously is that solar physics, radiation codes, ozone, and vertical resolution are often inadequately represented, making it challenging to effectively capture solar pathways, which are primarily top-down. Additionally, as the authors themselves note, reanalysis simulations are closely nudged to observations, so it is not surprising that the different ensemble members yield similar results at the surface, where nudging is strongest; this effect weakens in the upper atmosphere.

The authors provide a lengthy table with web addresses and details about the variables they analyzed, but they do not include crucial information regarding the underlying model physics, such as solar forcing, radiation, ozone, and nudged surface data. Having this information would be valuable for assessing the different results of the reanalysis, particularly in determining which model shows the most "realistic" response to solar influences and which aspects of the model physics are relevant.

Gonzalez et al. use an established multiple linear regression (MLR) approach to evaluate their observational and reanalysis datasets, expressing confidence in their results. However, they later state that MLR approaches are sub-optimal. This raises the question: why use them at all? Why not explore non-linear regression or a new machine learning approach? This inconsistency is difficult to understand. Furthermore, other approaches, such as the index method outlined by Ma et al. (2018), are not explained, leaving their application unclear.

Additionally, the calculation of significance levels is not adequately detailed, and I find the lack of a reasonable statistical analysis of the signal spread (variance analysis) concerning. Most importantly, there appears to be no analysis for the upper atmosphere or at least extending from the the tropopause to the surface. It would be essential to analyze whether solar signals penetrate, as this is critical for establishing confidence in the ocean signals? This analysis is absolutely necessary in my opinion.

**Results, Interpretation and Discussion**
Much of the presented results, appear similar to findings from previous studies, particularly those centered around the HadSLP dataset. The plots are often difficult to read, and different ranges for the color bars (and colors) are used for the same physical variable. Additionally, all plots are presented as single images, even though they are labeled as a, b, c. This should be unified for clarity. I also find it difficult to present 3 levels of significance in one plot. They are hard to distinguish from each other.

At the beginning, the authors promise to present a new mechanism that forms a new hypothesis. It would be helpful if the manuscript provided more detail and clarity on the proposed new mechanism and hypothesis. The main result, as I understand it, is illustrated in Figure 9, where the authors claim that a recurring anomaly in the deeper ocean reinforces the surface signal, suggesting a bottom-up controlled mechanism. I see it differently; I observe a surface signal that strengthens over time and then penetrates to deeper ocean layers, which is not particularly surprising. A thorough analysis of ocean dynamics is needed to support the authors point, as it currently relies on only a single plot, unfortunately.

The summary at the end again covers mostly already published results. The new findings, as explained, are not supported by sufficient analysis. It is interesting to see the authors suggest that the results may be related to teleconnections, such as Pacific-North Atlantic wave trains. If this had been demonstrated, it would indeed have been an exciting and novel result.

Chiodo, G., Oehrlein, J., Polvani, L. M., Fyfe, J. C., & Smith, A. K. (2019). Insignificant influence of the 11-year solar cycle on the North Atlantic Oscillation. Nature Geoscience, 12(2), 94-99.

Spiegl, T. C., Langematz, U., Pohlmann, H., & Kröger, J. (2023). A critical evaluation of decadal solar cycle imprints in the MiKlip historical ensemble simulations. Weather and Climate Dynamics, 4(3), 789-807.

Huo, W., Drews, A., Martin, T., & Wahl, S. (2024a). Impacts of North Atlantic model biases on natural decadal climate variability. Journal of Geophysical Research: Atmospheres, 129(4), e2023JD039778.

Huo, W., Spiegl, T., Wahl, S., Matthes, K., Langematz, U., Pohlmann, H., & Kröger, J. (2024b). Assessment of the 11-year solar cycle signals in the middle atmosphere in multiple-model ensemble simulations. EGUsphere.